# Association between Lower-Body Strength, Health-Related Quality of Life, Depression Status and BMI in the Elderly Women with Depression

**DOI:** 10.3390/ijerph19063262

**Published:** 2022-03-10

**Authors:** Carmen Galán-Arroyo, Damián Pereira-Payo, Ángel Denche-Zamorano, Miguel A. Hernández-Mocholí, Eugenio Merellano-Navarro, Jorge Pérez-Gómez, Jorge Rojo-Ramos, José Carmelo Adsuar

**Affiliations:** 1Promoting a Healthy Society Research Group (PHeSO), Faculty of Sport Sciences, University of Extremadura, 10003 Cáceres, Spain; magaar04@alumnos.unex.es (C.G.-A.); andeza04@alumnos.unex.es (Á.D.-Z.); jadssal@unex.es (J.C.A.); 2Health Economy Motricity and Education (HEME), Faculty of Sport Science, University of Extremadura, 10003 Cáceres, Spain; dpereirab@alumnos.unex.es; 3Physical Activity and Quality of Life Research Group (AFYCAV), Faculty of Sport Science, University of Extremadura, 10003 Cáceres, Spain; mhmocholi@unex.es; 4Grupo de Investigación EFISAL, Universidad Autónoma de Chile, Talca 3460000, Chile; emerellano@gmail.com; 5Social Impact and Innovation in Health (InHEALTH), Faculty of Sport Sciences, University of Extremadura, 10003 Cáceres, Spain

**Keywords:** depression, older women, lower-body strength, health-related quality of life, body mass index, geriatric depression scale, stand-up test

## Abstract

Introduction: Depression is currently the most prevalent mental illness in the world. It is a very frequent geriatric syndrome that causes a high degree of disability and increases mortality in the elderly population. This mental disorder is a social and public health problem that alters the quality of life (Qol) of the patient. Physical strength work has been reported to improve the clinical picture of people with depression. Objective. To determine the relationship between lower body strength, health-related quality of life (HRQoL), depression status and body mass index (BMI) in older women with depression. Design: A cross-sectional descriptive study with 685 elder women with depression. Results: A mild direct correlation (Rho = −0.29; *p* ≤ 0.001) between stand-ip test and EQ-5D-3L test was shown. There is a mild inverse correlation between stand-up test and six of fifteen items of the Geriatric Depression Scale (GDS) (Rho item 1 = −0.24; *p* ≤ 0.001; item 4 = 0.11; *p* ≤ 0.001; item 5 = −0.20; *p* ≤ 0.001; item 7 = −0.15; *p* ≤ 0.001; item 11 = −0.19; *p* ≤ 0.001; item 13 = −0.21; *p* ≤ 0.001). Between Stand-Up test and BMI, the correlation is weak inverse (Rho = −0.20; *p* ≤ 0.001). Conclusions: There is a significant association of lower body strength with HRQoL, and BMI, as well as some variables of depression status in elder women with depression. Better scores in the stand-up test lead to an improvement in HRQoL and BMI. Therefore, stand-up test could be a complementary tool in public health for improve HRQoL in the elderly women with depression.

## 1. Introduction

Depression is a high prevalence disease worldwide [1]. It is a fairly common syndrome in older adults [2], causing high disability and increasing mortality, in a direct and indirect manner due to comorbidities with other diseases in the elderly population [3]. It alters the QoL of the sufferer and is a social and public health problem. Depression is presented in different states in the elderly (minor depression, late-onset depression) [4], where physical and/or cognitive symptoms predominate. This mental disorder is characterized by persistent apathy and sadness, sleep disturbance and lack of appetite (WHO, 2017), causing low self-esteem, lack of concentration, and fatigue [5].

According to the WHO, 2017, it is estimated that, in the population over 55 to 79 years old, 7–8% of women suffer from depression. In Spain, according to the latest Spanish national health survey (ENSE, 2017), almost 10% of people over 60 years of age have depression.

Antidepressant therapy in the elderly presents some difficulties due to comorbidity with other chronic diseases [6], polypharmacy and increased sensitivity to the effects of drugs [7], and the fact that it is underdiagnosed [6,8].

Exercise has been shown to have major benefits that markedly improve the symptoms of geriatric depression [9]: better mood, improved self-esteem, improved Body Mass Index (BMI), and increased strength and functional capacity, which results in an improved QoL of the sufferer [10].

Conversely, depression decreases QoL [11], defined as the individual’s perception of their place in life in the cultural context and value systems in which they live and in relation to their goals, expectations, standards and concerns [12]. It is a very broad concept influenced by the individual’s physical health, psychological state, level of independence and relationship with the essential social and economic elements of the environment [13]. To assess QoL in a pathological population questions about symptom severity, daily functioning and other participants dimensions of well-being need to be included, with the goal of gaining a full understanding of a patient’s overall condition [14,15]. EQ-5D-3L test is one of the most commonly used instruments for this purpose [16].

In addition, depression is also negatively associated with strength levels [14], it decreases physical performance [17,18] and functional capacity in the elderly population [19]. Functional capacity is defined as an individual’s ability to perform activities of daily living without supervision needed [20]. Depression is a potential risk factor for frailty, falls, and disability [17]. One of the most widely used tests to evaluate lower body strength is the stand-up test, which belongs to the battery of physical fitness tests for older people by Rikli and Jones [21]. It assesses the power of the lower extremity muscles in a quicker and easier manner than an isokinetic dynamometer, without the need for special equipment or training. It consists in recording the number of times of standing and sitting in a chair for 30 s. Jones et al. [9] reported a higher reliability for the “30-s chair standing test” than for a previous test that consists in recording the time that an individual needs to stand a total of 5 or 10 times. This test is a way of functionally evaluating older adults because it measures lower body strength related to daily activities. In another study, it can be seen that older people with depression have lower scores in the stand-up test [22,23].

In relation to BMI, people with depression are at greater risk of being overweight or obese [24]. An association between depressive symptoms and increased body weight has been observed [25], and the other way around, being overweight or obese is associated with higher probabilities of suffering depression, which may be due to low self-esteem [26].

The assessment of depressive disorders symptoms in the elderly requires the use of rapid instruments with acceptable psychometric properties that allow an effective diagnostic approach for specialized personnel [27]. We measured depression levels in the elderly using the abbreviated Yesevage Geriatric Depression Scale (GDS) [28], which is specific to seniors population, that gives us information about self-perceived depressive state and quantifies depressive symptoms in older adults [27].

There are epidemiological studies on the possible factors that affect depression in the elderly [29,30]. Most agree that there is a link between depression and chronic diseases [31], which leads to a decrease in the QoL. These comorbidities, together with the antidepressant drug treatment, entail a polypharmacy that, in addition to causing a high cost for public health services, also leads to an increase in BMI, cognitive impairment [32,33], is associated with a poorer QoL [34] and lack of physical activity [35]; leading to a loss of strength, causing a sedentary lifestyle, increased BMI levels, low self-esteem and worse QoL, which negatively affect the state of depression.

After this contextual framing, knowing the relationship between lower body strength with HRQoL, depression status and BMI in older women with depression would have great interest.

## 2. Materials and Methods

### 2.1. Design

This is a descriptive cross-sectional study whose objective is to analyze the relationship between lower body strength through the stand-up test, HRQoL through the EQ-5D-3L, depression status through GDS and BMI in depressed elderly women.

The protocol was accepted by the Bioethics and Biosafety Committee of the University of Extremadura, in accordance with the ethical standards of the Declaration of Helsinki and national legislation on bioethics, biomedical research, sample confidentiality and data protection (117//2021).

Participants were informed about the procedures and signed an informed consent document before the start.

### 2.2. Sample

To know the minimum sample size, we assumed an alpha risk of 0.05 and a beta risk of 0.20, in bilateral contrast, the result of which was that at least 85 older women with depression were needed, accepting a correlation coefficient of 0.30 (moderate correlation according to Cohen’s classification [36]).

### 2.3. Participants

The participants were 685 older women with depression from the public health program “Exercise Look After You” (ELAY) of the Junta de Extremadura (Extremadura, Spain, 2019). The inclusion criteria for the study were being: women diagnosed with depression, who were over 59 years of age and who signed the informed consent form.

### 2.4. Procedures

#### 2.4.1. Stand-Up Test 30 s

It is an easy test to understand and perform, especially in older people. The procedure as described by Rikkli and Jones is as follows [21]: participants start the test sat with their arms crossed, close to the body at chest height (touching the opposite shoulders with their hands) and the back should be touching the back of the chair. The feet should be hip-width apart and touching the floor. At the start of the test, participants should stand-up and sit down as many times as they can within 30 s [21]. The reliability and validity of this test have been demonstrated in the elderly Spanish population [37].

#### 2.4.2. EQ-5D-3L

EQ-5D-3L is one of the most widely used instruments for measuring HRQoL [38]. It is a self-perceived health questionnaire. It consists of five health dimensions: (1) mobility; (2) self-care; (3) usual activities; (4) pain/discomfort; and (5) anxiety/depression, each with three levels of problems (none, a little or a lot) and a visual analogy scale, in which the individual self-rates his/her own health status from 0 (worst imaginable health) to 100 (best imaginable health) [39]. It has demonstrated validity and reliability in elderly people [40].

#### 2.4.3. GDS-15

It is designed for the screening of depression in the elderly. It is an abbreviated scale of 15 questions with dichotomous answers (yes/no). The result is the sum of the items, ranging from 0 (no depression) to 15 (maximum depression) points. Five points is the cut-off point that indicates depression. The Spanish version, GDS-SV, has shown reliability and validity in the elderly [41,42].

#### 2.4.4. BMI

The Body Mass Index (BMI) was calculated using the following equation: BMI = weight [kg]/(height [m]^2^).

### 2.5. Statistical Analysis

The data, collected by the technicians, were analyzed with the Statistical Package for the Social Sciences (SPSS) version 23.0 for MAC (IBM Corporation, Armonk, NY, USA).

First, the Kolmogorov–Smirnov test was performed to determine whether the data followed a normal distribution. The data did not follow a normal distribution, so it was decided to use nonparametric tests. Statistical significance was established at *p* < 0.05.

The relationship between the scores obtained in the EQ-5L-3D, the GDS and BMI, with respect to the standing test was obtained using Spearman’s Rho test.

As mentioned above, to interpret the correlation coefficient, Cohen’s classification ranges [36] <0.30 would be mild; 0.30 to 0.59 moderate; 0.6 to 0.79 high and ≥0.8 excellent were followed.

## 3. Results

Table 1 shows the Spearman’s correlation coefficients between stand-up test and EQ-5D-3L test. A mild direct correlation was demonstrated between the stand-up and the EQ-5D-3L index. There are also mild direct relationships in the Mobility, Self-care, Usual Activities and Pain/Discomfort items.

Table 2 shows the Spearman correlation coefficients between Stand-up and GDS, and the 15 items that comprise it. There is a mild inverse correlation between the stand-up value and the GDS items 1, 5, 7, 11, 13. Additionally, a mild direct relationship between stand-up and GDS4. For the rest of the items, the correlation was not significant.

Table 3 shows Spearman’s correlation coefficients between stand-up and BMI. There is a mild inverse correlation between stand-up and BMI.

## 4. Discussion

The main finding of this study is the significant association between lower-body strength (Stand-up) with HRQoL (EQ-5D-3L), depression status (GDS-SV-15) and BMI in older women with depression. Although it has been investigated in other populations with different diseases before [43], to our knowledge, this is the first study to relate lower body strength to these variables in older women with depression.

The results showed a mild direct relationship (Rho= −0.29; *p* ≤ 0.001) between stand-up and EQ-5D-3L, which means that older women with depression who obtain a higher physical performance in lower body strength would have a better HRQoL. Although there are no specific studies with depressives, there are previous works that relate lower body strength with QoL in the elderly [44,45,46] and in different diseases or pathologies [43,47,48]. There is no consensus about the relationship between stand-up and EQ-5D-5L. Some authors [49] had similar results to the present study, finding a small direct correlation between stand-up and each dimension of the EQ-5D-3L in older adults. Other authors [50] found a moderate-high correlation instead. It is generally accepted that strength improves the QoL of older people [49,51,52].

The tables presented in this manuscript show that there is a significant inverse association between different GDS variables and lower body strength [53,54,55]. This means that higher levels of lower body strength mean lower depression levels. Precisely, the GDS items where this association is stronger are the items more related to mood, which are *p* < 0.001: Are you satisfied with your life? Do you often feel bored? Do you feel happy most of the time? At this time, do you think it is great to be alive? Do you feel full of energy? This is in line with the findings of a recent study in Japanese elderly people revealing that depressed mood is associated with low muscle strength and physical performance [18]. Previous research has shown that higher physical performance is associated with lower levels of depression [53,54,55,56].

Regarding BMI, the results showed a small inverse correlation between stand-up and BMI (Rho = −0.20; *p* ≤ 0.001). This suggests that participants with a higher BMI will record fewer repetitions in the stand-up. Or, in other words, people with better fitness would have lower BMI. The existing evidence supports the association between obesity and adverse health outcomes among people with depression [57,58,59]. In addition, it has been found that higher BMI is related to a greater degree of body image dissatisfaction, which is related to an increase in depressive disorders [60].

Depression severely limits the daily activities of the elderly and drastically reduces their QoL [61,62], making it predisposing factor for poor QoL [63]. It is suggested that depressive symptoms are predictors of lower physical performance [17]. Additionally, poor physical performance predicts the future onset of depression [64], therefore, we could make use of this instrument (stand-up test) for the control of depression in the elderly.

### 4.1. Clinical Implications

If the significant relationships between the stand-up test and HRQoL, level of depression and BMI are confirmed, public health could use this tool as a complementary test for the early detection of depression, using it in their policies for the prevention of depressive disorders, as it is a low-cost instrument, easy and quick to apply and does not require any specialized equipment.

### 4.2. Limitations

A number of limitations must be taken into account in this study. It is cross-sectional, due to the COVID-19 pandemic situation, so we have had to conduct a correlational study, where cause–effect relationships are not established. This could be remedied in future studies with other designs, once the pandemic is over. Another limitation is that although the participants were referred from primary care services with a diagnosis of depression, due to the data protection law, we do not know the criteria followed to diagnose the illness, as well as the age of onset and comorbidities of the participants, so some important covariates may not have been adequately controlled. On the other hand, as all participants were diagnosed with depression, scores on the GDS screening tool scale were high, above 5 (the cut-off point above which depression is considered to be present), which may have limited the correlations found between lower body strength and GDS scale total score, as well as on most items. In addition, the study was only conducted in females; male participants and non-binary participants were not considered, because in the public health program from which the sample was drawn, 98% is female population and the number of males is not significant. Therefore, future consideration should be given to including men in the sample size to obtain sufficient statistical power to analyze the data by gender.

## 5. Conclusions

There is a statistical significant association of lower body strength with HRQoL, some items of GDS and BMI, as well as some variables of depression status in older women with depression. Better scores in the stand-up test lead to an improvement in HRQoL, BMI and some items of GDS test. Therefore, stand-up test could be a complementary tool in public health for improve HRQoL in elderly women with depression.

## Figures and Tables

**Table 1 ijerph-19-03262-t001:** Correlation between stand-up test and EQ-5D-3L in older women with depression.

Target Variable	Stand-Up			
	Spearman’s Rho	95% CI		*p*
Utility index	0.29	0.22	0.36	<0.001
EQ1: Mobility	−0.33	−0.40	−0.26	<0.001
EQ2: Self-care	−0.31	−0.38	−0.24	<0.001
EQ3: Usual Activities	−0.28	−0.35	−0.21	<0.001
EQ4: Pain/Discomfort	−0.22	−0.29	−0.15	<0.001
EQ5: Anxiety/Depression	−0.07	−0.14	0.00	0.053
VAS: Visual Analogic Scale	−0.02	−0.10	0.04	0.507

**Table 2 ijerph-19-03262-t002:** Correlation between stand-up test and GDS in older women with depression.

Target Variable	Stand-Up			
	Spearman’s Rho	95% CI		*p*
Geriatrics Depression Scale	−0.06	−0.13	0.01	0.099
1. In general, are you satisfied with your life?	−0.24	−0.30	−0.16	<0.001
2. Have you given up many of your usual tasks and hobbies?	−0.02	−0.09	0.05	0.585
3. Do you feel that your life is empty?	0.01	−0.05	0.09	0.684
4. Do you often feel bored?	0.11	−0.03	0.18	0.004
5. Are you in a good mood most of the time?	−0.20	−0.27	−0.12	<0.001
6. Are you afraid that something bad might happen to you?	0.02	−0.05	0.09	0.533
7. Do you feel happy most of the time?	−0.15	−0.22	−0.07	<0.001
8. Do you often feel helpless, unprotected?	0.05	−0.01	0.13	0.127
9. Do you prefer to stay at home rather than go out and do new things?	−0.05	−0.13	0.01	0.138
10. Do you think you have more memory problems than most people?	0.08	0.01	0.16	0.023
11. At this time, do you think it is great to be alive?	−0.19	−0.26	−0.12	<0.001
12. Do you currently feel useless?	<0.01	−0.07	0.07	0.902
13. Do you feel full of energy?	−0.21	−0.28	−0.14	<0.001
14. Do you feel hopeless at this time?	0.07	0.00	0.15	0.044
15. Do you feel that most people are better off than you?	<−0.01	0.07	0.07	0.938

**Table 3 ijerph-19-03262-t003:** Correlation between stand-up test and BMI in older women with depression.

Target Variable	Stand-Up			
	Spearman’s Rho	95% CI		*p*
Body Max Index (kg/m^2^)	−0.20	−0.27	−0.13	<0.001

## Data Availability

Data available upon request due to ethical and privacy restrictions. Data are not publicly available due to ethical and privacy restrictions. Data presented in this study are available upon request from the corresponding author.

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
