# Peer review of "Association between Lower-Body Strength, Health-Related Quality of Life, Depression Status and BMI in the Elderly Women with Depression"

_ijerph, 2022, doi:10.3390/ijerph19063262_

Round 1

Reviewer 1 Report

In this study, the authors have utilized a stand-up test to monitor depression status in older women. This cross-sectional study also used EQ-5D-3L and GDS-15 questionnaires to evaluate depression and BMI for health status. Overall the theme of this study is sound, but it lacks many details. The manuscript is poorly written, the methods are unclear, and the discussion needs more details.

Major comments

  • It is unknown in this study if the normal female of the same age with no history of any neuropsychiatric symptoms also show any abnormality in the stand-up test?
  • What was the inclusion and exclusion criteria? Do authors exclude any of data?
  • Why were only women included in the study?
  • It is important to discuss how depression was diagnosed?
  • It is very difficult to understand how the stand-up test was performed? It is suggested to provide a short video in the supplementary data to show the demonstration.
  • Statistical analysis in the results and/ or tables should include confidence intervals.

Minor comments

  • The word “quality of life” is repeatedly used is introduction and discussion section.
  • Flow of sentences in discussion section should be improved.

Author Response

First of all, thanks a lot for your contribution to improve our manuscript

In this study, the authors have utilized a stand-up test to monitor depression status in older women. This cross-sectional study also used EQ-5D-3L and GDS-15 questionnaires to evaluate depression and BMI for health status. Overall the theme of this study is sound, but it lacks many details. The manuscript is poorly written, the methods are unclear, and the discussion needs more details.

Major comments

  • It is unknown in this study if the normal female of the same age with no history of any neuropsychiatric symptoms also show any abnormality in the stand-up test?

In other research, we found significant differences between people with depression performing two fewer repetitions on the stand up test than people who did not have depression. as we have added in the introduction.

  • What was the inclusion and exclusion criteria? Do authors exclude any of data?

Participants were recruited from the public health program ELAY of the Junta de Extremadura, whose inclusion criteria are: people over 59 years of age, referred from primary care, living in Extremadura and able to perform a battery of health-related physical fitness tests.

For this research we added the following inclusion criteria: women over 59 years of age, referred with a diagnosis of depression by the Extremadura health service. Therefore, other pathologies and men were excluded.

  • Why were only women included in the study?

Because 98% of the people participating in the program are women. And the sample of men was not representative for conducting research. We added this data within the limitations to improve in future research.

  • It is important to discuss how depression was diagnosed?

We do not know the criteria followed by the health professional to diagnose the users of the program. Users are referred by primary care health specialists with a diagnosis of depression. We cannot access these data due to patient confidentiality. This is one of the limitations cited in the study.

  • It is very difficult to understand how the stand-up test was performed? It is suggested to provide a short video in the supplementary data to show the demonstration.

The description of the test protocol is defined in the article by Rikkli & Jones, which we have already mentioned, but we have adapted the paragraph to make it easier to understand. There you can see what the test consists of, it is very well defined. The description of the test protocol is defined in the article by Rikkli & Jones, which we have already mentioned, but we have adapted the paragraph to make it easier to understand. There you can see what the test consists of, it is very well defined.

  • Statistical analysis in the results and/ or tables should include confidence intervals.

 Modificied tables and confidence intervals included.

Minor comments

  • The word “quality of life” is repeatedly used is introduction and discussion section.

Thank you for your contribution, it is already modified.

  • Flow of sentences in discussion section should be improved.

The discussion has been improved.

Reviewer 2 Report

Dear Authors, This is an important issue as you correctly state.  Therefore this additional data may be of value to other researchers, clinicians and healthcare managers.  However, the abstract does not link the concepts mentioned in a coherent way.  The English used in the abstract does not match the standard of the main text. Please address these comments to improve the presentation of your results.  

1. Use a native English speaker who understands this research area to revise the abstract and the main text of the manuscript (see point 5).  There are sentences that are hard to understand and somewhat misleading.  For example: 

Abstract "It has been related that strength work improves the clinical picture of people with depression." Replace the word related with reported and insert the word physical in front of strength.   

2. Explain in the abstract how the stand-up test "controls" depression. Alternatively change the text to reflect the real use of the tool for screening the elderly patients for poor strength.  

3. Explain in your manuscript why your study was chosen as a cross-sectional design, when your introduction (lines 56 - 58) and abstract point out that an intervention of physical training has already been identified as a way to improve the mental health of the study group. 

4. Lines 97, 98 "After this contextual framing, it is interesting to know the relationship between lower body strength with (HRQoL), depression status and BMI in older women with depression.  Interesting confirmation, yes, but what is the new element to be gained? Explain fully in the introduction how this study could add to the established information and how it could inform clinical practice.  

5. Correct the sentences that ascribe agency to inanimate objects.  for example:  Lines 128, 129 "This test has demonstrated its reliability and validity in the Spanish elderly population [35]. "  The reliability and validity of this test have been demonstrated in the elderly Spanish population. 

5. Whole manuscript. Please replace the word subject with patient or participant. 

Best wishes

Author Response

Dear Authors, This is an important issue as you correctly state.  Therefore this additional data may be of value to other researchers, clinicians and healthcare managers.  However, the abstract does not link the concepts mentioned in a coherent way.  The English used in the abstract does not match the standard of the main text. Please address these comments to improve the presentation of your results.  

First of all, thanks a lot for your contribution to improve our manuscript.

  1. Use a native English speaker who understands this research area to revise the abstract and the main text of the manuscript (see point 5). 

Thank you for your input. The manuscript has been reviewed by a native English speaker.

There are sentences that are hard to understand and somewhat misleading.  For example: 

Abstract "It has been related that strength work improves the clinical picture of people with depression." Replace the word related with reported and insert the word physical in front of strength.   

Thanks, it has been replaced.

  1. Explain in the abstract how the stand-up test "controls" depression. Alternatively change the text to reflect the real use of the tool for screening the elderly patients for poor strength.

Thanks, it has been modified. We have explained in the abstract why the stand-up test can be a useful tool in public health for the control of depression in the elderly.

  1. Explain in your manuscript why your study was chosen as a cross-sectional design, when your introduction (lines 56 - 58) and abstract point out that an intervention of physical training has already been identified as a way to improve the mental health of the study group. 

It has been explained in the manuscript, in the limitations section.

  1. Lines 97, 98 "After this contextual framing, it is interesting to know the relationship between lower body strength with (HRQoL), depression status and BMI in older women with depression.  Interesting confirmation, yes, but what is the new element to be gained? Explain fully in the introduction how this study could add to the established information and how it could inform clinical practice.  

To our knowledge, this is the first study to relate lower body strength to these variables in older women with depression.

  1. Correct the sentences that ascribe agency to inanimate objects.  for example:  Lines 128, 129 "This test has demonstrated its reliability and validity in the Spanish elderly population [35]. " The reliability and validity of this test have been demonstrated in the elderly Spanish population. 

Thanks, it has been modified.

  1. Whole manuscript. Please replace the word subject with patient or participant. 

It have been modified.

Reviewer 3 Report

In this manuscript, the authors investigated the association between stand-up test performance and three variables, health-related QOL, depression, BMI in older women with depression. The study targets older women with depression, which has some merit. The manuscript, however, has several important issues to address.

1), the title is inconsistent with the research question in which the authors aimed to look at the relation between the stand-up test and three other variables, the reviewer cannot see why the stand-up test is useful for the "management of depression in the elderly".

2), the authors should, in the introduction, at least briefly describe how the stand-up test is conducted in order for the readers to understand the context of the study.

3), the authors did a priori power analysis to estimate the sample size required, however, they required ten times as many subjects. Isn't this overpowered and a waste of resources?

4), how was depression diagnosed in the public health program? according to DSM-4 or 5, for instance? Did the authors collect information on the comorbidity of depression? and onset age?

5), in section 2.5 statistical analysis, the authors stated that the data did not follow a normal distribution, did they mean all the variables that they investigated?

6), the main statistical analysis used in this study is merely spearman correlation, which is oversimplified. Multivariate analysis controlling the influence of covariates such as age, onset age, comorbidity, etc, is better conducted.

7), the authors failed to detect a correlation between the stand-up test and GDS score and then focused on item-level correlations. These item-level correlations are hardly meaningful and the lack of a significant correlation between the stand-up test and the total score is somewhat inconsistent with the studies that the authors introduced. Relevant to comment 6), the authors may try to control potential confounding factors to identify a pure correlation. Another method better than the item-level correlation is perhaps conducting principal component analysis or factor analysis to identify latent variables of GDS and then investigate the correlation between these latent variables and the stand-up test.

8), for data clarify, the authors may provide a scatterplot showing all the individual variations for the main correlations.

Author Response

In this manuscript, the authors investigated the association between stand-up test performance and three variables, health-related QOL, depression, BMI in older women with depression. The study targets older women with depression, which has some merit. The manuscript, however, has several important issues to address.

First of all, thanks a lot for your contribution to improve our manuscript

1), the title is inconsistent with the research question in which the authors aimed to look at the relation between the stand-up test and three other variables, the reviewer cannot see why the stand-up test is useful for the "management of depression in the elderly".

Thank you for your input, following your indications, it has already been modified so that the tittle seems more accurate to the findings

2), the authors should, in the introduction, at least briefly describe how the stand-up test is conducted in order for the readers to understand the context of the study.

Thank you, it was added

3), the authors did a priori power analysis to estimate the sample size required, however, they required ten times as many subjects. Isn't this overpowered and a waste of resources?

You are right, but in this case we have added more participants because they were part of the ELAY program and there was no waste of time because the data were already there.

4), how was depression diagnosed in the public health program? according to DSM-4 or 5, for instance? Did the authors collect information on the comorbidity of depression? and onset age?

We do not know how depression was diagnosed. That type of data is not treated for patient confidentiality. All we know is that they come to the ELAY program with a medical diagnosis of depression.

5), in section 2.5 statistical analysis, the authors stated that the data did not follow a normal distribution, did they mean all the variables that they investigated?

Precisely that. The data did not follow a normal distribution for any of the variables, so it was decided to use non-parametric tests.

6), the main statistical analysis used in this study is merely spearman correlation, which is oversimplified. Multivariate analysis controlling the influence of covariates such as age, onset age, comorbidity, etc, is better conducted.             7), the authors failed to detect a correlation between the stand-up test and GDS score and then focused on item-level correlations. These item-level correlations are hardly meaningful and the lack of a significant correlation between the stand-up test and the total score is somewhat inconsistent with the studies that the authors introduced. Relevant to comment 6), the authors may try to control potential confounding factors to identify a pure correlation. Another method better than the item-level correlation is perhaps conducting principal component analysis or factor analysis to identify latent variables of GDS and then investigate the correlation between these latent variables and the stand-up test.

We have performed a regression introducing different variables that could be important to explain depressive symptoms such as age, MCI or comorbidity with diabetes. A table with the linear regression models is attached.

We believe that most of the variables were not significant in terms of the homogeneity of the sample data on these variables (e.g. of the 685 depressive persons, 592 were diabetic).

8), for data clarify, the authors may provide a scatterplot showing all the individual variations for the main correlations.

Thank you very much for your suggestion. We think that being dichotomous variables the graphs do not contribute much to our study.

Round 2

Reviewer 1 Report

The authors have addressed all my comments, and the manuscript is now improved. I recommend to accept this manuscript in its current form. I detect only one minor typo in the title of the manuscript “Association Of Health-related Quality of Life, Depression and BMI In The Elderly Women”

Author Response

We appreciate your words, as well as your contributions to improve our manuscript.

Reviewer 3 Report

Unfortunately, most of the reviewer's concerns, including 1), the diagnosis criteria and comorbidity, etc, 2), item-level correlation is hardly meaningful, 3) important covariates are not adequately controlled, 4), scatterplots are not provided, etc., have not been adequately addressed. Without addressing these issues, the reviewer is not sure about the contribution of this work and does not recommend the acceptance of this manuscript.

Author Response

In line with the editor's recommendations, we have included their concerns in the limitations section. Thank you.